# On the Melting Thresholds of Semiconductors under Nanosecond Pulse Laser Irradiation

Jiří Beránek [1,2], Alexander V. Bulgakov [1] and Nadezhda M. Bulgakova [1,*]

1  HiLASE Centre, Institute of Physics of the Czech Academy of Sciences, Za Radnicí 828, 25241 Dolní Břežany, Czech Republic
2  Faculty of Nuclear Sciences and Physical Engineering, Czech Technical University in Prague, Trojanova 13, 12001 Prague, Czech Republic
*  Correspondence: bulgakova@fzu.cz

**Abstract:** In this work, a unified numerical model is used to determine the melting thresholds and to investigate the early stages of melting of several crystalline semiconductors (Si, Ge, GaAs, CdTe and InP) irradiated by nanosecond laser pulses. A molten fraction approach is used for continuous transition over the melting point. The results are compared with previously published theoretical and experimental data. A survey on the thermophysical and optical properties of the selected materials has been carried out to gather the most relevant data on temperature dependent properties for the solid and liquid states of these semiconductors where such data are available. A generalization of the obtained results is established that enables evaluation of the melting thresholds for different semiconductors based on their properties and irradiation conditions (laser wavelength, pulse duration).

**Keywords:** pulsed laser; nanosecond; laser processing; semiconductors; melting; thermal model; finite difference method; material properties

## 1. Introduction

The material processing of semiconductors using short and ultrashort laser pulses is one of the key technologies in various fields, including microelectronics, photonics, photovoltaics, sensor devices. It has been employed for enhancing dopant diffusion [1], crystallization [2] and selective modification of multilayer structures [3], and in studies of kinetics of structural changes in materials [4]. Its main advantages are the high level of controllability and variety of wavelengths that can be selected to fit a particular material and an application. One of the basic parameters for laser material processing that involves surface patterning, modification, crystallization and ablation, is the melting (damage) threshold. This key parameter is usually determined in the experimental studies of laser material processing as a reference point for controlling the laser modification process and for testing theoretical models developed with the aim of better understanding the fundamental processes at laser–matter interaction [5–20]. For detection of the phase change, several approaches are used, such as time-resolved reflectivity (TRR) [9,11,19–22], acoustic [23] and electrical conductivity [24] measurements and pump-probe microscopy [25], or combined techniques as in [13], where time-of-flight velocity distributions together with the evaporation rate and reflectivity were measured and analyzed.

In some cases, the experiments identify a transient state where only a minor or instable change of the studied parameter is observed and thus the obtained melting threshold, i.e., minimal laser energy density (laser fluence), needed to reach this state corresponds to the melting onset. In other cases, a stable change of the measured value is detected, corresponding to the established molten phase on the irradiated surface at higher fluences. Thus, in TRR experiments, the obtained data on the reflectivity of the molten phase and melting threshold are strongly dependent on the sensitivity and time resolution of the

measurements [19] and on the wavelength of the probe laser beam [20]. For instance, a UV beam, which probes a thinner surface layer, results in a lower melting threshold of silicon as compared to a visible probe beam with a higher absorption depth under identical conditions [20]. We believe, therefore, that the threshold values obtained with the TRR technique are generally overestimated if we assume the onset of melting as a threshold. It should also be noted that the fluence interval with a transient melting is dependent on the material purity and surface quality such as roughness, which can lead to a locally changed absorption, as well as on the presence of hotspots in the laser beam profile. In modeling, it is typically assumed that the melting threshold corresponds to the energy density needed for the temperature of the surface of the material to reach the melting point, $T_m$ [5,6,26–28].

A comprehensive analytical theory has previously been developed for optical heating of semiconductors using the coupled diffusion equations for heat transfer and excited carrier density and taking into account a variety of non-linear processes involved, such as free-carrier and two-photon absorption, radiative and Auger recombination and temperature shift of the band gap [26]. The theory was applied to evaluate the melting thresholds of a number of semiconductors irradiated by laser pulses in a range of pulse durations from nanoseconds to a millisecond, and in some cases a reasonable agreement with available experimental data was obtained [27,28]. However, the practical use of this theory is rather complicated as it requires data on a large number of physical properties of the material, including those related to free carrier dynamics and their dependencies on the temperature and carrier density, which have to be analytically integrated over laser-heating time. As a result, the accuracy of this approach to predict the melting thresholds is relatively low, especially for short pulse durations, below ~30 ns. Furthermore, the theory does not consider the molten state of materials. On the other hand, numerical simulations are nowadays very efficiently and rather routinely used in the field of laser–matter interaction and can be a valuable alternative to the sophisticated analytical approach.

In the presented work, we have used the classical heat transport model to investigate the laser-induced heating and melting of several semiconductors (Si, Ge, GaAs, CdTe, InP) irradiated by single laser pulses at wavelengths from 248 to 694 nm in the range of pulse durations from 7 to 70 ns. The novelty of our approach lies primarily in the application of the same unified model to a variety of materials and irradiation conditions without any adjustable free parameters. We compare results obtained in our modeling with available experimental data and theoretical predictions on the damage (melting) thresholds and characteristics of the melting stage (its duration and melting depth). We assume that the experimentally observed damage of semiconductors is due to melting of the surface. Our model enables one to follow the melting dynamics, including the evolution of the melt fraction. The results obtained for different semiconductors have been generalized in order to predict their melting thresholds based on a unified parameter combining irradiation conditions and material optical and thermophysical properties.

## 2. Model Description

The thermal model is applicable for nanosecond laser pulse durations and longer pulses because the electron–lattice interaction phenomena, critical for shorter pulses, are not accounted for, and the coupling of laser energy to the material lattice is treated as an instantaneous local process [29]. This is justified by the fact that electron–lattice thermalization time is in the order of ~$10^{-12}$ s for silicon and germanium [7], and similar time scales for the other materials under study. In addition, as we consider relatively low surface temperatures, below and near the melting point, we disregard evaporation phenomena, which, however, may slightly affect the melting process for compound semiconductors [30,31]. For correct simulations of laser–matter interaction processes, material thermophysical and optical properties (and their temperature dependences) are of fundamental importance. The material parameters used in the presented calculations are summarized in Appendix A, Tables A1–A19.

The sample is assumed to be flat, and the laser beam couples perpendicularly to the surface in the direction of the $z$ axis. The simulations are considered as a one-dimensional (1D) problem that is a valid approximation, as long as the irradiation spot size (typically above 100 µm for the considered experiments) is much larger than the absorption depth, as is our case. Indeed, for a ruby laser with the longest wavelength in this study, the absorption depth of the studied semiconductors is less than 0.4 µm (see Appendix A). For shorter wavelengths, also investigated here, the absorption depths are even smaller. Therefore, the time-dependent temperature distribution in the irradiated target is governed by the heat-flow equation in its 1D form [32,33]:

$$(c_{\mathrm{p}}(T)\rho + L_{\mathrm{m}}\delta(T - T_{\mathrm{m}}))\frac{\partial T}{\partial t} = \frac{\partial}{\partial z}\left(\kappa(T)\frac{dT}{dz}\right) + S(z, t). \tag{1}$$

Here, $t$ is time, $T$ is the temperature and $c_{\mathrm{p}}$, $\rho$, $L_{\mathrm{m}}$, $T_{\mathrm{m}}$ and $\kappa$ are, respectively, the heat capacity, the density, the latent heat of fusion, the melting temperature and the thermal conductivity of the sample material. Energy supplied by the laser is represented by the source term $S(z, t)$ as:

$$S(z, t) = (1 - R)I(t)\alpha \exp(-\alpha z) \tag{2}$$

where $R$ and $\alpha$ are the surface reflectivity and the material absorption coefficient. The pulse intensity $I(t)$ has a Gaussian temporal profile:

$$I(t) = \frac{2F_0}{\tau_{\mathrm{L}}\sqrt{\frac{\ln 2}{\pi}}}\exp\left(-4\ln 2\left(\frac{t}{\tau_{\mathrm{L}}}\right)^2\right), \tag{3}$$

with $F_0$ and $\tau_{\mathrm{L}}$ being the peak fluence and the pulse duration.

Equation (1) is solved numerically using the finite difference method and the implicit scheme that ensures a high numerical stability. For temperatures below the melting point, the finite difference form of Equation (1) is written on the numerical grid as:

$$-\kappa_{\mathrm{l}}T_{i-1}^* + \left(\frac{\Delta z^2}{\Delta t}\rho c_{\mathrm{p}} + \kappa_{\mathrm{l}} + \kappa_{\mathrm{r}}\right)T_i^* - \kappa_{\mathrm{r}}T_{i+1}^* = \frac{\Delta z^2}{\Delta t}\rho c_{\mathrm{p}}T_i + S(z, t), \tag{4}$$

where

$$\kappa_{\mathrm{l}} = \frac{\kappa_{i-1} + \kappa_i}{2}, \ \kappa_{\mathrm{r}} = \frac{\kappa_{i+1} + \kappa_i}{2} \tag{5}$$

and index $i$ refers to the numerical grid points. The temperature values, $T^*$, are unknown at the time moment $t_{\mathrm{f}}$, and $T$ (without asterisk) corresponds to the known temperature at the time moment $t_{\mathrm{f}-1} = t_{\mathrm{f}} - \Delta t$.

One of the advantages of the implicit numerical scheme is that using large spatial and/or temporal steps ($\Delta z$ and $\Delta t$, respectively) does not affect its stability; however, too large $\Delta t$ values can introduce truncation errors to the calculation results [34]. Similarly, the choice of $\Delta z$ should enable a good approximation of the laser intensity attenuation toward the material depth and the temperature gradient within the heat affected zone. A very good approximation was achieved with $\Delta t$ values of 5–10 ps and $\Delta z$ = 1 nm. The sample is considered to be semi-infinite. The system of the linear Equation (1) with discretization to the form (4) represents a tridiagonal matrix, which is solved by the Thomas algorithm [35].

Material heating to the melting point followed by the melting process leads to an accumulation of the internal energy at constant $T = T_{\mathrm{m}}$, and its ratio to the enthalpy of melting can be interpreted as a molten fraction in a computational element. In the presented calculations, we apply the method of through calculation without explicit selection of the phase interface [32,33]. According to this method, the melting process is smoothed over a symmetric interval of a width of a few Kelvins around the melting point. Melting starts at a slightly lower temperature than $T_{\mathrm{m}}$, reaches the melting point at the fraction of molten material of 0.5, and ends at a slightly higher temperature than $T_{\mathrm{m}}$. In the interval of melting,

a $\delta$ function is added to the heat capacity term to account for absorption/release of the fusion heat at the melting/solidification front:

$$\delta(T) = \frac{L_\mathrm{m}}{A\sqrt{\pi}} e^{-\left(\frac{T-T_\mathrm{m}}{A}\right)^2},$$

(6)

where $A$ is the width of the delta function in Kelvin. As the internal energy rises upon laser light absorption, the physical parameters are gradually changing from the solid to the liquid phase proportionally to the fraction of molten material [36]:

$$\gamma = \gamma^\mathrm{s}(T)(1-\eta) + \eta\gamma^\mathrm{l}(T),$$

(7)

where $\gamma_\mathrm{s}$, $\gamma_\mathrm{l}$ represent a property of material in solid and liquid state, respectively, and $\eta$ is the fraction of molten material. For the sake of simplicity, the change in density upon melting is not taken into account so that the value of the solid-state density is also kept for the liquid state.

In the presented model, we interpret the fluence needed for the fraction of molten material to reach the interval from 0–3% as the melting threshold fluence, $F_\mathrm{th}$. According to the δ-function approach (Equations (1) and (6)), this occurs at ~1–2 K bellow the tabulated melting point and thus the edge of beginning of melting is blurred. However, from analyzing our simulation data, it follows that the position of $F_\mathrm{th}$ within the interval of melting has a minor effect on its resulting value. Furthermore, taking into account the ambiguity of $F_\mathrm{th}$ reported in the literature, this aspect plays only a small role.

## 3. Results and Discussion

Here we present an analysis of the available literature data used for our model development and discuss the simulation results and general trends in the damage threshold determination. The results of the present simulations are summarized in Table 1 in comparison with the literature data. In addition, for irradiation conditions with a ruby laser (wavelength 694 nm) where the most systematic data are available, we have performed calculations for various laser pulse duration of 15, 30 and 70 ns, beyond the ranges reported in the literature, in order to investigate the effect of pulse duration for specific materials (the obtained results are also presented in Table 1).

**Table 1.** The simulation results for the damage threshold fluence, $F_\mathrm{th}$, of the studied semiconductors in comparison with theoretical and experimental data reported in the literature. The experimental data are marked by asterisks *.

| Material | $\lambda$, nm | $\tau$, ns | $F_\mathrm{th}$, mJ/cm$^2$ This Work | $F_\mathrm{th}$, mJ/cm$^2$ Literature Data |
|---|---|---|---|---|
| Si | 532 | 18 | 355 | 395 [6], 330 * [37] |
|  |  | 30 | 423 | 474 [6] 350 [38] |
|  | 694 | 15 | 672 | 725 [6] |
|  |  | 30 | 752 | 805 [6], 980 [27] |
|  |  | 70 | 900 |  |
| Ge | 694 | 15 | 191 |  |
|  |  | 30 | 255 | 300 [27] |
|  |  | 70 | 370 | 400 * [10] |
| GaAs | 308 | 30 | 213 | 200, 200 * [8] |
|  | 532 | 15 | 184 |  |
|  | 694 | 15 | 265 | 300 [12] |
|  |  | 20 | 282 | 250 * [13], 360 * [23], 240 [27] |
|  |  | 30 | 316 |  |
|  |  | 70 | 415 |  |

**Table 1.** *Cont.*

| Material | $\lambda$, nm | $\tau$, ns | $F_{th}$, mJ/cm$^2$ This Work | $F_{th}$, mJ/cm$^2$ Literature Data |
|---|---|---|---|---|
| CdTe | 248 | 20 | 46 | 50, 50 * [39] |
| | 694 | 15 | 68 | 78 [28] |
| | | 30 | 80 | 98 [28] |
| | | 70 | 103 | 130 [28] |
| InP | 532 | 7 | 106 | 97 [30] |
| | 694 | 15 | 165 | |
| | | 30 | 211 | |
| | | 70 | 296 | |

Note that, in the literature, the $F_{th}$ values can be determined differently from the method used in this study. For example, in Ref. [39], the calculated melting threshold for CdTe was set 8% higher than the laser fluence needed for reaching $T_m$. Time resolved reflectometry (TRR) measurements performed by the authors did not show an increase in reflectivity at the intensity corresponding to reaching the melting point in the calculations. Thus, as the melting threshold, the authors consider the intensity at which the sample surface layer is molten to the depth of laser radiation absorption. Experimentally measured values of the melting threshold fluence typically include a transition interval where localized melting occurs, giving rise to an increase in the reflectivity above the values of solid-state surface reflectivity [13,29]. In numerical simulations, the determination of the damage threshold strongly depends on the used material properties [40]. Their choice can be considered as the main source of inaccuracy of the modeling results because we use a very accurately verified implicit algorithm for numerical solution (Section 2). Below we have surveyed the literature for the optical and thermophysical parameters of the studied semiconductors. The most relevant parameters are given in Appendix A.

*3.1. Silicon*

Some ambiguity exists in the reported melting point of crystalline Si, ranging from 1683 to 1690 K [7,29]. For the temperature dependence of $c_p$, we took the data from Ref. [41] with a stronger variation over the range of solid-state temperatures than the dependence used for *c*-Si in Ref. [5]. The *c*-Si thermal conductivity was approximated by the expression from the measured data reported in [42]. For the liquid state, we use $c_p$ = 910 J/(kg·K) and $\kappa = 50.8 + 0.029(T - T_m)$ W/(m·K) [5]. The reflectivity and the absorption coefficient for *c*-Si are temperature dependent and given by the relations presented in [43]. The optical properties of molten silicon are described according to the calculated data for 694 nm [44] and measured data for 352 nm [45]. The data on the properties for solid and liquid silicon used in the present modeling are summarized in Tables A1–A4 of Appendix A.

Interestingly, our simulation data for Si (Table 1) somewhat overestimate the melting threshold fluence measured in [37,38], while they are systematically lower than the simulated $F_{th}$ values presented in [6]. The experimental investigations reported in [37] for 532 nm ns laser irradiation give an interval of increasing reflectivity between 330 and 380 mJ/cm$^2$. The value of 380 mJ/cm$^2$ was identified as a threshold for reaching a high reflectivity (~70%), probably indicating melting to a depth of approximately one optical skin layer (~10 nm), and the maximum reflectivity of 73% was observed at 450 mJ/cm$^2$. A similar value of around 400 mJ/cm$^2$ was determined as a threshold for melting based on time-resolved reflectivity measurements [38]. The theoretical calculations [6] give higher values for the melting thresholds than those calculated in this work and measured in [37,38]. The main reason for this can be seen in the difference in the absorption coefficient change with temperature. For instance, the absorption coefficient of *c*-Si at a 694 nm wavelength at temperatures close to the melting point is approximately five times larger in our case (taken from [43]) than in calculations presented in [6]. Note that the analytical theory [27] predicts an even higher threshold fluence for Si under the same irradiation

conditions (694 nm, 30 ns, see Table 1). As a whole, our modeling data for Si are in reasonable agreement with most of the published data, thus demonstrating that our model approach can be used for other semiconducting materials. The calculations with various pulse durations, $\tau_L$, show that the melting threshold increases with $\tau_L$ proportionally to app $\tau_L^{0.2}$ (Table 1), i.e., the dependence is considerably weaker than the $\propto \tau_L^{0.5}$ dependence predicted for the *evaporation* threshold for fairly long (nanosecond and longer) laser pulses [46].

### 3.2. Germanium

The next set of simulations has been carried out for germanium for the conditions of the experiments reported in [10]: wavelength 694 nm and pulse duration 70 ns. Time-resolved reflectivity measurements using a probe 1.06 μm laser wavelength identified the energy density of 400 mJ/cm² as a value, at which the rise of reflectivity was detected corresponding to the observable melting. Numerical simulations using the finite difference method were also carried out in Ref. [10], and the obtained melting threshold was claimed to be "practically identical" to the measured one (although the method for threshold determination in the simulation was not specified). The authors used experimentally measured values of reflectivity and absorption from Refs. [9,47], which are in good agreement with the optical constants we derived from measurements reported in [48] and also confirmed in [49]. Our calculations give $F_{th}$ = 370 mJ/cm² (Figure 1a, Table 1), which is in good agreement with the data [10], particularly taking into account that the increase in reflectivity detected in [10] assumes a significant fraction of molten germanium and thus a slightly higher fluence than that needed to reach the melting temperature at the surface. Near the melting threshold, the calculated melt fraction reaches a maximum after a delay of approximately 20 ns relative to the moment of laser peak intensity (Figure 1a), which is also in agreement with the measurements [10]. With the known properties of liquid Ge (Tables A7 and A8), we have performed simulations for $F > F_{th}$, which are again in good agreement with the measured durations of a high reflectivity stage corresponding to molten germanium [10] (Figure 1b). It should be mentioned that different values are reported for the thermal conductivity of liquid Ge. In the modeling, we use the value of 29.7 W/(m·K) [50], while in Ref. [51], $\kappa$ = 43 W/(m·K) was measured. The calculations performed for various pulse durations demonstrate a stronger $\tau_L$ dependence than that for silicon, close to the $\propto \tau_L^{0.5}$ dependence (Table 1).

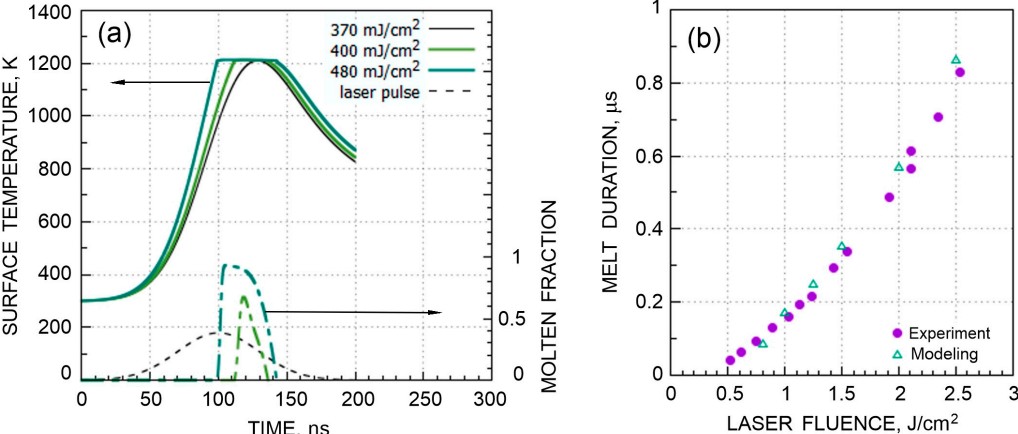

**Figure 1.** (**a**) Surface temperature (solid lines) and molten fraction (dot-dashed lines) of germanium obtained in the modelling for different laser fluences (694-nm, 70-ns pulse). The laser temporal profile is shown by the dashed line. (**b**) Comparison of the duration of increased reflectivity (data from experiment [10]) and the melt duration obtained in our simulations.

### 3.3. Gallium Arsenide

For simulations of the laser heating of GaAs, we used the same values of the thermophysical properties as in Ref. [8]. For the temperature dependence of $c_p$, the data from [52] were used, which are also in a good agreement with the data reported in [53]. Optical properties were taken from measurements [54], which are also in a good agreement with [55]. The absorption and reflection coefficients were calculated from the refractive index and the extinction coefficient and taken as temperature-independent. The material properties used in the simulations are presented in Tables A9–A12 of Appendix A.

Several regimes of laser irradiation of GaAs corresponding to available experimental and theoretical data were investigated in our modelling, with the laser wavelengths ranging from 308 to 694 nm and pulse duration ranging from 15 to 70 ns. For all the conditions, the melting thresholds calculated here with our unified model are in good agreement with the values reported in the literature (Table 1). Below we discuss each irradiation regime in more detail.

*λ = 308 nm, τ = 30 ns.* Our model implements the same optical and temperature-dependent material properties as in the model presented by Kim et al. [8]. For the solid-state reflectivity and the optical absorption, the data used for simulations in [8] are in agreement with the measured data for solid GaAs [53]. As the parameters of the model [8] and ours are very similar, we take this comparison as a validation for our model that gives a deviation of only ~6% (see Table 1).

*λ = 694 nm, τ = 15 ns.* García et al. [12] carried out simulations for a ruby laser with a 15 ns pulse duration using an explicit numerical scheme. The melting threshold was identified at a laser fluence of 300 mJ/cm$^2$, which corresponded to the situation when a ~65-nm-thick surface layer was molten [12]. In our simulations, this fluence of 300 mJ/cm$^2$ results in a melting depth of 13 nm, while the melting threshold corresponding to reaching the melting point on the sample surface is 265 mJ/cm$^2$ (see Figure 2 for comparison). The difference in the melting depth can be attributed to two factors. First, the authors [12] extrapolated the temperature-dependent absorption coefficient for the solid-state GaAs from the room temperature util the melting point, which appears to be questionable. In our simulations, we use constant but reliable data on the optical absorption and reflectivity of molten GaAs at the wavelength of the ruby laser [48]. The reflectivity coefficient of liquid GaAs in both Ref. [12] and this work was adopted from [13], $R = 0.67$. The second factor may be related to using an explicit numerical scheme in Ref. [12], whose approximation to the initial equations often represent a challenge.

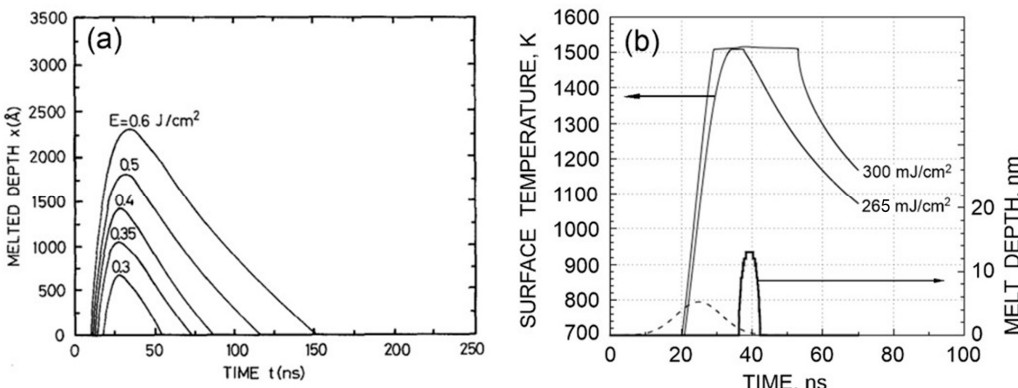

**Figure 2.** Comparison of the results obtained by modelling in Ref. [12] and in this work for GaAs irradiated by 694-nm, 15-ns laser pulses. (**a**) The depth of molten material as a function of time for several laser fluences (reprinted with permission from Ref. [12]). (**b**) The results of the present modeling for laser fluences of 265 mJ/cm$^2$ (corresponding to the defined melting threshold) and 300 mJ/cm$^2$. The temporal evolution of the melt depth at 300 mJ/cm$^2$ is also given to compare with Ref. [12]. The laser pulse is shown by the dashed line.

$\lambda = 694\ nm$, $\tau = 20\ ns$. Pospieszczyk et al. [13] presented two sets of measurements. Using a HeNe probe laser, the temperature-dependent reflectivity was investigated. The second set of data gives time-of-flight measurements of particles evaporated from the GaAs surface (Figure 3a). Comparison of their experimental data and our simulations is given in Table 1, which are in reasonable agreement. The simulated damage threshold associated with achieving the melting temperature (Figure 3b) is somewhat higher than in the experiments [13] but is still in the range of fluences where a transient uneven melting is observed (Figure 3). This discrepancy, although relatively small, can be related to the effect of decreasing the melting temperature due to depletion of the target surface by a more volatile component [30,31,56], which is not taken into account in our model.

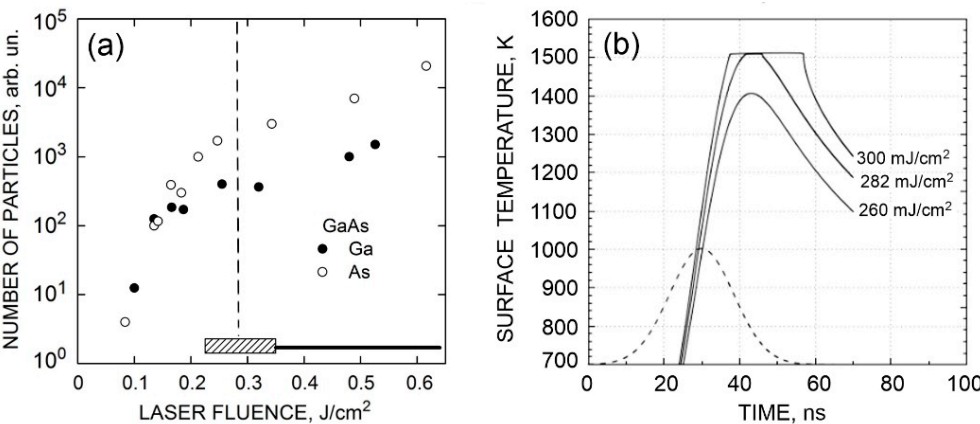

**Figure 3.** (**a**) The number of Ga and As atoms emitted from the GaAs surface irradiated by 694-nm, 20-ns laser pulses as a function of laser fluence as derived from mass spectrometric measurements (adapted with permission from Ref. [13]). Transient uneven and developed manifestations of increased reflectivity indicating the appearance of the liquid phase are marked by the shaded region and the solid line, respectively. Our simulated melting threshold is shown by a vertical dashed line. (**b**) The simulated dynamics of the surface temperature with the identified melting threshold of 282 mJ/cm$^2$. The laser pulse profile is shown by the dashed line.

The TRR measurements [23] for these conditions (694 nm, 20 ns) gave a considerably higher threshold value for GaAs (Table 1), likely due the fact that the low sensitivity of the measurements required establishing a well-developed melting stage to register the change in reflectivity (the authors of [23] estimated a melt thickness of ~20 nm at the threshold fluence). The theoretical prediction [28] is in good agreement with our simulations and measurements [13] in this case (Table 1).

### 3.4. Cadmium Telluride

We have applied our model to CdTe irradiated by a KrF excimer laser (248 nm) for the conditions of Gnatyuk et al. [37], where TRR measurements and numerical simulations of the pulsed laser heating of CdTe were performed. For this material, reliable physical and optical properties are extensively reported in the literature. In our simulations, the value of thermal conductivity was taken from Refs. [57,58] for solid and liquid state, respectively. The specific heat for both solid and liquid state was taken from Ref. [59]. The same thermophysical properties were also used in simulations [31,37]. Measurements of the optical properties of CdTe using spectroscopic ellipsometry and modeling were performed in [60] for a wide range of wavelengths. Reflectivity and absorption are the same for solid and liquid state and independent of temperature.

The authors [37] identified a laser fluence of 50 mJ/cm$^2$ as the melting threshold. In their simulations, this value corresponds to the molten layer with a thickness of the laser absorption depth. Their TRR measurements detected an abrupt although small rise of the reflectivity at a laser fluence of 48–50 mJ/cm$^2$. These results are in excellent agreement

with our simulations (Table 1). Indeed, for $F = 50$ mJ/cm$^2$, our model gives the depth of the molten layer of 7 nm, very close to the absorption depth of CdTe at 248 nm (~9 nm). According to our definition of the melting threshold, achieving the melting temperature at the very surface of the irradiated sample, the calculated threshold is slightly lower, 46 mJ/cm$^2$ (Table 1).

Note that the analytical theory predictions [28] of the melting thresholds of CdTe at a laser wavelength of 694 nm are in reasonable agreement with our simulations over the studied range of pulse durations, although the theoretical values are systematically higher than ours (Table 1). We should also note that, in Refs. [32,56], an effect of the enhanced evaporation of Cd atoms enriching the surface by tellurium upon laser heating was studied. It was shown that this effect can have an impact on the melting and ablation processes. This effect was not taken into account in this work, nor in [28,37].

### 3.5. Indium Phosphide

We have applied our model for the conditions of experiments [30], where InP was irradiated by a nanosecond laser at $\lambda = 532$ nm. In this paper, a laser fluence of 97 mJ/cm$^2$ was identified as the damage threshold. In our simulations, we have obtained a threshold value of 106 mJ/cm$^2$, which can be considered as a good agreement taking into account that there are no fitting parameters in our model. It should be mentioned that, although laser processing of InP is a common technique in its industrial applications, the thermophysical parameters at enhanced temperatures are still not well studied. Thus, several sets of data are available for the heat capacity of solid InP; see, e.g., [61]. The major problem is that measurements of the thermophysical properties at enhanced temperatures are affected by the high vapor pressure of phosphorous due to its high volatility. The thermal conductivity and the specific heat of molten InP are given in [53]. The reflectivity and absorption are calculated from data provided in [48]. Optical properties are taken as temperature independent and are considered the same for both solid and liquid state. In reference article [30], ablation of compound semiconductors is studied, and a model that takes into account evaporation of their components gives the melting threshold. Our result, which disregards this effect, gives $F_{\text{th}}$ that is approximately 10% higher.

### 3.6. Generalization of the Damage Threshold Data into a Predictive Dependence

A wide set of data on the damage thresholds of five semiconductors under various ns-laser irradiation conditions are obtained in our calculations in the frames of a unified thermal model, and all the thresholds are in good agreement with available literature data, both experimental and theoretical ones. The obtained threshold values vary in a wide range depending on material, from ~50 mJ/cm$^2$ for CdTe to almost 1 J/cm$^2$ for Si (Table 1). The irradiation conditions also affect the threshold values, which are generally smaller for shorter laser wavelengths and pulse durations. It is very attractive to generalize the obtained results in terms of a unified parameter combining the basic material properties (thermophysical and optical) in order to be able to predict the ns-laser-induced melting thresholds, at least approximately, without performing detailed simulations.

Bäuerle [29] considered "optimal" melting conditions during ns-laser-induced thermal surface melting, when minimal laser energy is required for a certain melt depth. Assuming that such conditions are fulfilled when the melt depth is equal to the heat-diffusion length, he estimated the optimal laser fluence as:

$$P_{\text{B}} = \frac{2\rho \Delta H}{1 - R} \left( \frac{D}{\tau_{\text{L}}} \right)^{\frac{1}{2}} \tau_{\text{L}} \tag{8}$$

where $D = \kappa/\rho c_{\text{p}}$ is the thermal diffusivity and $\Delta H = L_{\text{m}} + c_{\text{p}}(T_{\text{m}} - 300)$ is the total energy needed to heat the sample to the complete melting state from room temperature, A similar parameter was introduced in [46] as an evaporation threshold under ns-laser ablation

(assuming naturally by $\Delta H$ in Equation (8) the specific heat for evaporation instead of that for melting and omitting the $2/(1 - R)$ factor).

Figure 4 shows the calculated melting threshold values plotted as a function of the $P_B$ parameter, Equation (8), evaluated for all the studied materials using their room-temperature properties. All the data are nicely grouped around a straight line in the logarithmic plot. This clear correlation is rather surprising for such a simplified generalization approach when the material absorption coefficient and temperature dependencies of thermophysical properties are not taken into account. The least square fitting line in Figure 4 is described by a power law, $F_{th} \approx 0.05\, P_B^{1.16}$, which can be used for a rough estimation of the melting threshold of semiconductors based on their basic room-temperature properties.

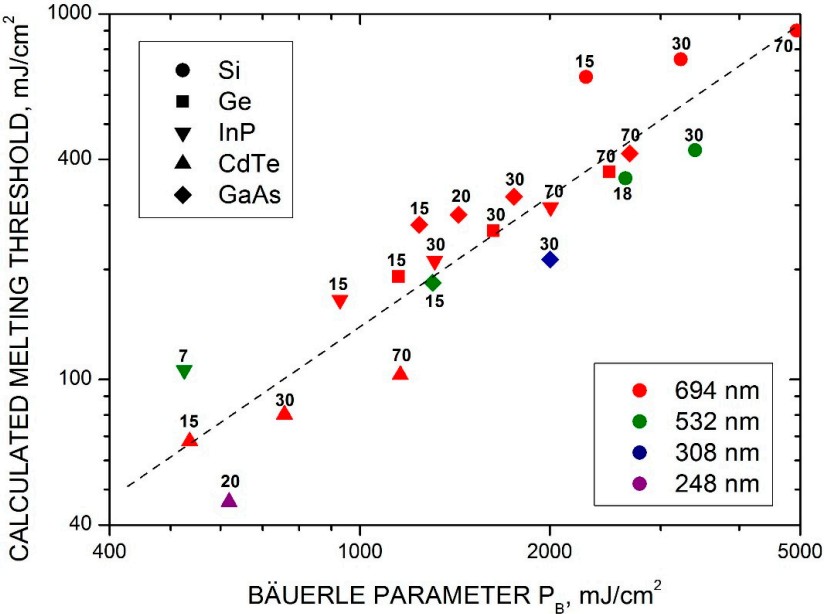

**Figure 4.** Calculated melting thresholds of the studied semiconductors for different laser wavelengths as a function of the Bäuerle parameter, $P_B$, Equation (8). The numbers above the points correspond to the laser pulse duration in nanoseconds. The line represents a power-law least-square fit of the data.

The parameter $P_B$ predicts a growth of the melting threshold with the laser pulse duration as $\tau_L^{0.5}$. However, as was noticed above, this is not always the case according to our simulations. Some semiconductors (Ge, CdTe, InP) follow closely the $\tau_L^{0.5}$ dependence, while others (Si, GaAs) demonstrate weaker dependencies (Table 1 and Figure 4). This finding agrees with calculations based on the theory in [26], which also predicts a nearly $\tau_L^{0.5}$ dependence for CdTe over a wide range of pulse durations [28] and a much weaker dependence for Si [27]. This is probably mainly due to a difference in the thermal diffusivity, $D$, of the materials. Thus, at room temperature, $D \approx 0.8\ \text{cm}^2/\text{s}$ for Si, and it is around $0.35\ \text{cm}^2/\text{s}$ for Ge, InP and CdTe. A higher thermal diffusivity results in a higher heat diffusion length and smaller in-depth temperature gradients and thus in a lower heat flow from the surface at an increased pulse duration. The temperature dependencies of material parameters (included in our model simulations) can additionally affect the pulse duration dependence of the melting threshold.

## 4. Conclusions

In this work, based on the classical thermal model, we have developed a numerical approach to investigate the continuous solid–liquid phase change in solid targets heated by nanosecond laser pulses. The model is applied to a number of semiconductors and various irradiation conditions, and the obtained results on the melting thresholds, melt duration and melt depth are compared with experimental and theoretical data

available in the literature. The comparison is not always straightforward as the value presented as melting threshold fluence is not always describing the same state of the studied material. However, in most cases, good agreement with the literature data is obtained. The simulations also predict the dependence of the melting thresholds on the laser pulse duration, which is found to be material dependent and weaker than that expected from simple heat-flow considerations. A good correlation of all the calculated melting threshold values with a parameter combining material thermophysical properties and surface reflectivity is obtained. The correlation can be used as a simple method for estimation of the melting thresholds of ns-laser irradiated semiconductors based on their room-temperature properties.

**Author Contributions:** Conceptualization, A.V.B. and N.M.B.; methodology, J.B. and N.M.B.; validation, J.B., A.V.B. and N.M.B.; formal analysis, A.V.B.; investigation, J.B.; writing—original draft preparation, J.B.; writing—review and editing, J.B., A.V.B. and N.M.B.; visualization, J.B., A.V.B. and N.M.B.; funding acquisition, N.M.B. All authors have read and agreed to the published version of the manuscript.

**Funding:** This work was supported by the European Regional Development Fund and the state budget of the Czech Republic (project BIATRI: No. CZ.02.1.01/0.0/0.0/15_003/0000445). J.B. acknowledges funding from the Grant Agency of the Czech Technical University in Prague, No. SGS22/182/OHK4/3T/14.

**Conflicts of Interest:** The authors declare no conflict of interest.

## Appendix A

Here we provide all the parameters for semiconductors, which were selected after a thorough literature analysis and used in our modeling. Some reliable data, which are widely cited in the literature and web-sites, are given without references.

### Silicon

**Table A1.** *c*-Si—thermophysical properties.

| Property | Value | Ref. |
|---|---|---|
| $\rho$, g/cm$^3$ | 2.328 | |
| $T_{\mathrm{m}}$, K | 1688 | |
| $L_m$, J/kg | $1.826 \times 10^6$ | [62] |
| $c_p$, J/kg K | $847.05 + 118.1 \times 10^{-3}\, T - 155.6 \times 10^5\, T^{-2}$ | [42] |
| $\kappa$, W/mK | $97269\, T^{-1.165}$ $(300 < T < 1000)$ $3.36 \times 10^{-5}\, T^2 - 9.59 \times 10^{-2}\, T + 92.25$ $(1000 < T < T_{\mathrm{m}})$ | [42] |

**Table A2.** *c*-Si—optical properties.

| Property | Value, 532 nm | Ref. | Value, 694 nm | Ref. |
|---|---|---|---|---|
| $n$ | 4.152 | [48] | 3.79 | [48] |
| $k$ | 0.051787 | [48] | 0.013 | [48] |
| $R$ | 0.374 | [43] | $0.34 + 5 \times 10^{-5}\,(T - 300)$ | [43] |
| $\alpha$, 1/m | $5.02 \times 10^5 \exp{(T/430)}$ | [43] | $1.34 \times 10^5 \exp{(T/427)}$ | [43] |

**Table A3.** *Liquid*-Si—thermophysical properties.

| Property | Value | Ref. |
|---|---|---|
| $\rho$, g/cm$^3$ | 2.52 | |
| $c_p$, J/kg K | 910 | [5] |
| $\kappa$, W/mK | $50.28 + 0.029(T - T_{\mathrm{m}})$ | [5] |

**Table A4.** *Liquid*-Si—optical properties.

| Property | Value, 532 nm | Ref. | Value, 694 nm | Ref. |
|:---:|:---:|:---:|:---:|:---:|
| $n$ | 3.212 | [45] | 3.952 | [44] |
| $k$ | 4.936 | [45] | 5.417 | [44] |
| $R$ | 0.693 | Calculated | 0.707 | Calculated |
| $\alpha$, $1/m$ | $1.1659 \times 10^8$ | Calculated | $9.804 \times 10^7$ | Calculated |

## Germanium

**Table A5.** *c*-Ge—thermophysical properties.

| Property | Value | Ref. |
|:---:|:---:|:---:|
| $\rho$, $g/cm^3$ | 5.327 | |
| $T_m$, K | 1211.4 | |
| $L_m$, J/kg | $5.1 \times 10^5$ | [50] |
| $c_p$, J/kg K | $1.17 \times 10^{-1} T + 293$ | [50] |
| $\kappa$, W/mK | $18{,}000/T$ | [50] |

**Table A6.** *c*-Ge—optical properties.

| Property | Value, 694 nm | Ref. |
|:---:|:---:|:---:|
| $n$ | 5.04 | [48] |
| $k$ | 0.49 | [48] |
| $R$ | 0.45 | Calculated |
| $\alpha$, $1/m$ | $8.81 \times 10^6$ | Calculated |

**Table A7.** *Liquid*-Ge—thermophysical properties.

| Property | Value | Ref. |
|:---:|:---:|:---:|
| $\rho$, $g/cm^3$ | 5.6 | |
| $T_m$, K | 3106 | |
| $c_p$, J/kg K | 450 | [50] |
| $\kappa$, W/mK | 29.7 | [50] |

**Table A8.** *Liquid*-Ge—optical properties.

| Property | Value, 694 nm | Ref. |
|:---:|:---:|:---:|
| $n$ | 2.62 | [44] |
| $k$ | 5.238 | [44] |
| $R$ | 0.74 | Calculated |
| $\alpha$, $1/m$ | $9.485 \times 10^7$ | Calculated |

## Gallium Arsenide

**Table A9.** *c*-GaAs—thermophysical properties.

| Property | Value | Ref. |
|:---:|:---:|:---:|
| $\rho$, $g/cm^3$ | 5.32 | |
| $T_m$, K | 1511 | |
| $L_m$, J/kg | $7.11 \times 10^5$ | |
| $c_p$, J/kg K | $8.76 \times 10^{-2} T + 308.16$ | [8] |
| $\kappa$, W/mK | $30{,}890\, T^{-1.141}$ | [8] |

**Table A10.** *c*-GaAs—optical properties.

| Property | Value, 308 nm | Ref. | Value, 532 nm | Ref. | Value, 694 nm | Ref. |
|---|---|---|---|---|---|---|
| $n$ | 3.7 | [48] | 4.13 | [48] | 3.78 | [48] |
| $k$ | 1.9 | [48] | 0.336 | [48] | 0.15 | [48] |
| $R$ | 0.42 | Calculated | 0.37 | Calculated | 0.338 | Calculated |
| $\alpha$, 1/m | $7.7 \times 10^7$ | Calculated | $8.04 \times 10^6$ | Calculated | $2.687 \times 10^6$ | Calculated |

**Table A11.** *Liquid*-GaAs—thermophysical properties.

| Property | Value | Ref. |
|---|---|---|
| $c_p$, J/kg K | 439.85 | [8] |
| $\kappa$, W/mK | $30{,}890\ T^{-1.141}$ | [8] |

**Table A12.** *Liquid*-GaAs—optical properties.

| Property | Value, 308 nm | Ref. | Value, 694 nm | Ref. |
|---|---|---|---|---|
| $R$ | 0.46 | [8] | 0.67 | [13] |
| $\alpha$, 1/m | $0.83 \times 10^8$ | [63] | $2.687 \times 10^6$ | Taken the same as for solid-state |

## Cadmium Telluride

**Table A13.** *c*-CdTe—thermophysical properties.

| Property | Value | Ref. |
|---|---|---|
| $\rho$, g/cm$^3$ | 5.85 | |
| $T_{\mathrm{m}}$, K | 1365 | |
| $L_m$, J/kg | $2.09 \times 10^5$ | [59] |
| $c_p$, J/kg K | $3.6 \times 10^{-2}\ T + 205$ | [59] |
| $\kappa$, W/mK | $1507/T$ | [57] |

**Table A14.** *c*-CdTe—optical properties.

| Property | Value, 248 nm | Ref. | Value, 694 nm | Ref. |
|---|---|---|---|---|
| $n$ | 2.63 | [60] | 3.037 | [60] |
| $k$ | 2.13 | [60] | 0.286 | [60] |
| $R$ | 0.406 | Calculated | 0.258 | Calculated |
| $\alpha$, 1/m | $1.1 \times 10^8$ | Calculated | $5.179 \times 10^6$ | Calculated |

**Table A15.** *Liquid*-CdTe—thermophysical properties.

| Property | Value | Ref. |
|---|---|---|
| $\rho$, g/cm$^3$ | 6.4 | |
| $c_p$, J/kg K | 255 | [59] |
| $\kappa$, W/mK | 1.1 | [37] |

**Table A16.** *Liquid*-CdTe—optical properties.

| Property | Value, 248 nm | Ref. |
|---|---|---|
| $R$ | 0.45 | [37] |
| $\alpha$, 1/m | $1.1 \times 10^8$ | [37] |

## Indium Phosphide

**Table A17.** *c*-InP—thermophysical properties.

| Property | Value | Ref. |
|---|---|---|
| $\rho$, g/cm$^3$ | 4.81 | |
| $T_m$, K | 1335 | |
| $L_m$, J/kg | $3.4 \times 10^5$ | [64] |
| $c_p$, J/kg K | $2.33 \times 10^{-2}\,T + 347$ | [61] |
| $\kappa$, W/mK | $1.215 \times 10^5\,T^{-1.324}$ | [61] |

**Table A18.** *c*-InP—optical properties.

| Property | Value, 532 nm | Ref. | Value, 694 nm | Ref. |
|---|---|---|---|---|
| $n$ | 3.702 | [41] | 3.49 | [48] |
| $k$ | 0.429 | [41] | 0.27 | [48] |
| $R$ | 0.335 | Calculated | 0.31 | Calculated |
| $\alpha$, 1/m | $1.013 \times 10^6$ | Calculated | $4.82 \times 10^6$ | Calculated |

**Table A19.** *Liquid*-InP—thermophysical properties.

| Property | Value | Ref. |
|---|---|---|
| $c_p$, J/kg K | 424 | [53] |
| $\kappa$, W/mK | 22.8 | [53] |

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
