# Peer review of "On the Melting Thresholds of Semiconductors under Nanosecond Pulse Laser Irradiation"

_applsci, doi:10.3390/app13063818_

Round 1
Reviewer 1 Report
In my opinion, the paper is of good quality and can be accepted without modifications.
Author Response
We are very grateful for the high evaluation of our manuscript by the Reviewer. It is encouraging and we believe that our paper will also be appreciated by the community.
Reviewer 2 Report
The manuscript is well written and has all necessary sections The lit. review in the introduction section needs to be improved significantly. More recent works in the same field need to be cited and discussed. The novelty of this model is not still clear and there are experimental results to validate the theoretical results clearly. Source of errors and accuracy of the model results need to be added and discussed. In general , it is well written but need more works to be ready for publication.
Author Response
Response to Reviewer 2 Comments
We are grateful to the Reviewer for careful reading our manuscript and providing comments which are useful for its improvement.
Point 1: The lit. review in the introduction section needs to be improved significantly. More recent works in the same field need to be cited and discussed.
Response 1: We have updated the literature in Introduction, including several recent papers, and added discussions not only in Introduction but also in further sections. We have also added new literature data on the melting threshold to Table 1 to compare with our simulation results and discussed them in the corresponding sections.
Point 2: The novelty of this model is not still clear and there are experimental results to validate the theoretical results clearly.
Response 2: We have added to the Introduction the following sentence: “The novelty of our approach lies primarily in application of the same unified model to a variety of materials and irradiation conditions without any adjustable free parameters.” Other novelties were written in the initial version of the manuscript: a possibility to investigate the melting dynamics (melting stage duration, melting depth, molten fraction) as well as generalizing our modeling results using the Bäuerle parameter, which is very simple and enables an express evaluation of the damage threshold that can be then used for its more accurate determination via computer simulations.
Point 3: Source of errors and accuracy of the model results need to be added and discussed.
Response 3: : In the Section 2, we present a detailed description of our model and simulation algorithm with giving time and space steps at which we obtained excellent approximation (that is the temperature dynamics does not noticeably change upon further decreasing the time and/or space steps). It should be noted that the used here implicit algorithm has unconditional convergence that is highly beneficial for robust and fast computer simulations, contrary to explicit algorithms which are widely used for such kind of simulations.
Thus, the main source of errors lies in the choice of the material properties. In this work, we have thoroughly reviewed the related materials properties and summarized the most relevant parameters in Appendix. After a rigorous literature overview, we have chosen the parameters which were verified experimentally in further studies. In view of many material parameters included in our model, it is not possible to estimate the accuracy related to these parameters. For this, one would need to know “exact” material properties.
In the second paragraph of Section 2, we have added the following sentence: “Their choice can be considered as the main source of inaccuracy of the modeling results since we use a very accurately verified implicit algorithm for numerical solution (Section 2).”
We hope that we have convincingly responded to Reviewer’s remark and that our manuscript can be considered for publishing in Materials.

Author Response
Response to Reviewer 3 Comments
We are grateful to the Reviewer for useful questions and comments which allow to further improve our manuscript.
Point 1: How did the author define the damage threshold fluence?
Response 1: We are grateful for this important point. The damage threshold is considered as the melting threshold when first signs of the material modifications can be identified after solidification. We have added the following sentence in Introduction: “We assume that the experimentally observed damage of semiconductors is due to melting of the surface.”
Point 2: What is the laser spot size defined in the model? Does the spot size of the laser affect the melting threshold?
Response 2: In the initial version of the manuscript, we write: “The simulations are considered as a one-dimensional (1D) problem that is a valid approximation as long as the irradiation spot size (typically above 100 µm for the considered experiments) is much larger than the absorption depth that is our case.”
Indeed, according to the optical properties provided in Appendix, in all considered cases the absorption depth is below 1 micrometer and reduces with heating and upon melting. A violation of 1D approach can be expected for a super-tight focusing toward spot sizes of a few micrometers when the spot size can affect the melting process but such focusing regimes are practically not used for nanosecond laser pulses.
Point 3: The pulsed duration is one of the critical parameters to affect the melting threshold. The short pulsed duration will probably induce the non-linear absorption on the materials. Should we consider the non-linear effect in the model when the pulses are short?
Response 3: For non-linear absorption in the regimes around or even well higher that the melting threshold, it is necessary to have a high flux of photons (e.g. two-photon absorption is described by the term of alpha2×I2 vs. alpha1×I for single photon absorption). Usually, the corresponding constant alpha1 is many orders of magnitude higher than alpha2. This effect can be important for femtosecond laser pulses with terawatt intensities as was well analyzed by Sokolowski-Tinten and von der Linder (Phys. Rev. B 61, 2643 (2000)). Thus, we believe that single photon (intrinsic) absorption of semiconductors in the spectral range where they are opaque is dominating is dominating at nanosecond laser pulses and two-photon (and higher non-linear) absorption does not contribute noticeably to light absorption. Other nonlinear effects like Kerr nonlinearity do not also contribute to the light absorption at ns laser pulses as laser intensities used at ns laser ablation experiments is much lower than the intensity thresholds for nonlinear effects. Two-photon absorption is not contributing to material excitation even in the case of silicon damage by UV fs laser pulses (Bulgakova et al. Appl. Phys. A 81, 345 (2005)). Taking into account nonlinear optical effects for modeling of ns laser action on materials will only complicate the description with no impact on its results. Thus, we do not consider nonlinear effects in the frames of our manuscript.
Point 4: What is the repetition rate of the nanosecond laser used in this work? Will it be helpful to improve the simulation results if the authors consider the repetition rate of the laser in the developed model?
Response 4: All the simulations reported to this manuscript are performed for single pulse action. Correspondingly, we have compared the modeling results with single short laser damage in both available experiments and theoretical works of other authors. We have clarified this in Introduction:
“In the presented work, we have used the classical heat transport model to investigate laser-induced heating and melting of several semiconductors (Si, Ge, GaAs, CdTe, InP) irradiated by single laser pulses at wavelengths from 248 to 694 nm in the range of pulse durations from 7 to 70 ns.”
Point 5: Si and Ge share similar atomic structure. Have the author considered the relationship between the properties of the atomic structure and the melting threshold?
Response 5: The thermal model we use is macroscopic and, thus, it is based on the macroscopic material properties whose values are summarized in Appendix of our manuscript. Therefore, the atomic structure does not enter directly into the model.
Indeed, for molecular dynamics or more sophisticated quantum models like time-dependent density functional theory (e.g., Derrien et al. Phys. Rev. B 104, L241201 (2021)), the realistic atomic structure is one of the most important model features. However, such approaches are not yet applicable for nanosecond laser pulses as they are extremely time- and cost-demanding.

Round 2
Reviewer 2 Report
The revised manuscript can be accepted